# Analysis of the *AIRE* Gene Promoter in Patients Affected by Autoimmune Polyendocrine Syndromes

**DOI:** 10.3390/ijms25052656

**Published:** 2024-02-24

**Authors:** Annamaria Cudini, Caterina Nardella, Emanuele Bellacchio, Alessia Palma, Domenico Vittorio Delfino, Corrado Betterle, Marco Cappa, Alessandra Fierabracci

**Affiliations:** 1Bambino Gesù Children’s Hospital, IRCCS, 00165 Rome, Italy; annamaria.cudini@opbg.net (A.C.); caterina.nardella@opbg.net (C.N.); alessia.palma@opbg.net (A.P.); 2Molecular Genetics and Functional Genomics, Bambino Gesù Children’s Hospital, IRCCS, 00165 Rome, Italy; emanuele.bellacchio@opbg.net; 3Section of Pharmacology, Department of Medicine and Surgery, University of Perugia, 06129 Perugia, Italy; domenico.delfino@unipg.it; 4Padua University, 35128 Padua, Italy; corrado.betterle@unipd.it; 5Research Unit for Innovative Therapies in Endocrinopathies, Bambino Gesù Children’s Hospital, IRCCS, 00165 Rome, Italy; marco.cappa@opbg.net

**Keywords:** *AIRE*, *AIRE* gene promoter, polymorphisms, autoimmune polyglandular syndrome, autoimmune etiopathogenesis, sequencing

## Abstract

Autoimmune polyglandular syndromes (APS) are classified into four main categories, APS1–APS4. APS1 is caused by *AIRE* gene loss of function mutations, while the genetic background of the other APS remains to be clarified. Here, we investigated the potential association between *AIRE* gene promoter Single Nucleotide Polymorphisms (SNPs) and susceptibility to APS. We sequenced the *AIRE* gene promoter of 74 APS patients, also analyzing their clinical and autoantibody profile, and we further conducted molecular modeling studies on the identified SNPs. Overall, we found 6 SNPs (-230Y, -655R, -261M, -380S, -191M, -402S) of the *AIRE* promoter in patients’ DNA. Interestingly, folding free energy calculations highlighted that all identified SNPs, except for -261M, modify the stability of the nucleic acid structure. A rather similar percentage of APS3 and APS4 patients had polymorphisms in the *AIRE* promoter. Conversely, there was no association between APS2 and *AIRE* promoter polymorphisms. Further *AIRE* promoter SNPs were found in 4 out of 5 patients with APS1 clinical diagnosis that did not harbor *AIRE* loss of function mutations. We hypothesize that *AIRE* promoter polymorphisms could contribute to APS predisposition, although this should be validated through genetic screening in larger patient cohorts and in vitro and in vivo functional studies.

## 1. Introduction

A book published to celebrate the 63rd anniversary of the discovery of autoimmune disorders (AIDs) edited by the “fathers” of autoimmunity describes more than a hundred of AIDs and estimated to affect about 7% of the general population [1,2,3]. In their natural history, it is generally observed progression from latent, to subclinical toward clinical disease with associated disease-related circulating autoantibodies (Abs) [4,5]. Subsequently, criteria for their frequent association called autoimmune polyglandular syndrome (APS) [6], multiple autoimmune (MAS) [4,7], or polyautoimmunity syndromes [8] were established. Indeed, the association was not limited to polyglandular diseases but it can also include multiple organ-specific autoimmune disorders, such as endocrine, gastrointestinal, skin, neurologic, and non-organ-specific rheumatologic conditions. 

APS includes four main categories [4,5,6]. As regard APS1, also called autoimmune–polyendocrinopathy-candidiasis-ectodermal dystrophy syndrome (APECED, OMIM #240300, ORPHA 3453), is a rare monogenic recessive disorder caused by loss of function mutations in the AutoImmune REgulator (*AIRE*) gene, whose clinical diagnosis requires the presence of at least two of the following diseases: chronic mucocutaneous candidiasis (CMC), chronic hypoparathyroidism (HP) and primary adrenal insufficiency (Addison’s disease, AD) [9]. The combination of autoimmune thyroid disease (AITD), Type 1 diabetes mellitus, (T1DM), and AD is the autoimmune polyglandular syndrome Type 2 (APS2, Schmidt’s syndrome). This is a rare disease in humans, occurring in 1.4–4.5 per 100.000 inhabitants in Europe (OMIM #269200, ORPHA 3143). APS3 was defined as the association between AITD and one or more AIDs excluding AD. APS3 (ORPHA 227982) included four main subgroups based on the associated AITD: APS3A (AITD and autoimmune endocrine diseases), APS3B (AITD with autoimmune gastrointestinal, hepatic, or pancreatic diseases), APS3C (AITD with autoimmune skin, neurological and hematological diseases), APS3D (AITD with autoimmune rheumatological, cardiac, and vascular diseases). In the consideration that AITD is one of the most frequent autoimmune diseases and that about one-third of the patients can be associated with a non-thyroid AID during the entire lifespan, the APS3 can be considered the most prevalent APS worldwide [4,5,10,11]. APS4 (ORPHA 227990) is the last category including any other AID combination that cannot be assigned to APS1, APS2 or APS3 [4,5,6]. Overall, the incidence of APS2 to 4 is estimated between 1.4 and 4.5 per 100.000 inhabitants according to published studies [12]. 

Whereas the genetics of APS1 are clearly defined, APS2, APS3, and APS4 are genetically complex multifactorial syndromes [12]. The inheritance pattern seems to be autosomal-dominant with incomplete penetrance in some patients with several genetic loci being involved through interaction with environmental factors [13]. The cluster of several different organ-specific and non-organ-specific autoimmune diseases in patients can be due to shared common proinflammatory genetic background as well as a defect in immune regulation [14,15,16,17]. APS2, APS3, and APS4 are strongly associated with certain alleles of HLA genes of the major histocompatibility complex (MHC) located on chromosome 6. As regards the haplotypes, HLA-DRB1*03:01-DQA1*05:01-DQB1*02:01 and HLA-DRB1*04:01-DQA1*03:01-DQB1*03:02 are significantly overrepresented and therefore predisposing, whereas the HLA-DRB1*15:01-DQA1*01:02-DQB1*06:02 haplotype is underrepresented and therefore protective of the development of these APS. The presence of the DRB1*04-DQ8 haplotype differentiates between APS1 and APS2. The DRB1*04:04-DQA1*03:01-DQB1*03:02 haplotype is associated with APS2, whereas the DRB1*04:01-DQA1*03:01-DQB1*03:02 haplotype confers susceptibility for APS3 [12]. 

Furthermore, several gene variations in non-HLA genes are present in APS. Among these, the *PTPN22* (protein tyrosine phosphatase non-receptor type 22) C1858T variant encoding for the R620W Lyp (rs2476601) is frequently associated with T1DM, AITD, AD, and the APS2 syndrome [18]. Other gene polymorphisms associated with APS are detected in the *CTLA4* gene [19] encoding for the cytotoxic T lymphocyte-associated antigen-4, the vitamin D receptor (VDR) gene [20], the *IL2ra* gene encoding IL2Ra (CD25) [21], the *TNFα* (tumor necrosis factor alpha) gene [22,23], the *FOXP3* (forkhead box P3) gene which controls Treg development and function [24] and the MHC class I chain-related gene A (*MICA*) [25,26]. Further, susceptibility to T1DM is conferred by variable number of tandem repeats (VNTR) of the insulin gene [27,28]. It is generally recognized that genetic variability in the *AIRE* locus and the presence of heterozygous loss of function *AIRE* mutations can affect the presentation of self-antigens in the thymus and thus the development of certain organ-specific autoimmune disorders [29]. *AIRE* variants were detected in the DNA of patients affected by organ-specific autoimmune disorders [30]. *AIRE* gene monoallelic mutations located in the first plant homeodomain (PHD1) zinc finger with autosomal dominant inheritance were found associated with autoimmune disorders with later onset, milder phenotype, and reduced penetrance that did not satisfy the clinical diagnostic criteria for APECED [31]. In a recent paper by Oftedal et al. [32], 20 individuals from 11 kindreds with dominant *AIRE* mutations within the PHD1 e PHD2 domains were identified. These variants were shown to have dominant negative effects in vitro. 

In the light of the foregoing, since the expression profile of peripheral tissue antigens in the thymus could not only be affected by *AIRE* deficiency, in the present study, we aimed to investigate whether the susceptibility to APS could be instead affected by SNPs of the *AIRE* gene promoter [33], potentially inducing alteration of the *AIRE* gene transcription.

## 2. Results 

### 2.1. Clinical Phenotype and AIRE Gene Promoter Screening

As shown in Table 1, several *AIRE* gene promoter polymorphisms were identified in APS patients and conventionally numbered with respect to the *AIRE* start codon, which was assigned the value 0. These included the heterozygous -230Y (C/T) SNP (rs751032) (Appendix A) and the homozygous -230T SNP, whose genotype frequencies were similar to those found in healthy controls (HD) (Table 2) [34]. Five additional heterozygous SNPs were rarely detected in five different APS patients: the -655R (G/A) (rs117557896), the -261M (C/A) (rs934375604), the -380S (C/G) (rs371261300), the -191M (C/A) (rs1048356976), and the -402S (C/G). The latter was not previously reported in genome databases (Appendix A). Heterozygous polymorphisms -655R (G/A), -452Y (C/T) (rs547103905), -214M (A/C) (rs184978263), and -124M (C/A) (rs868650327) were detected in normal controls (Table 2B) (Appendix A). Regarding the frequency of haplotypes -655G -230Y (C/T) and -655G -230T, there were no differences between patients and HD groups. Particularly, 19 patients and 18 HD presented haplotype -655G -230Y, while 2 patients and 3 HD showed haplotype -655G -230T (Table 2).

No specific correlation was observed with peculiar serum autoantibody specificities and the presence of polymorphisms of *AIRE* gene promoter.

In Appendix A, the *AIRE* gene pattern of each patient is also reported. *AIRE* gene polymorphisms IVS9+6 G>A (c.1095+6 G>A, rs1800525) and S278R (c.834 C>G, rs1800520), previously associated with APS [35], were identified in 17 and in 14 out of 74 patients, respectively. No significant association was found between the presence of these *AIRE* gene polymorphisms and the described *AIRE* gene promoter variants. As can be seen, among 17 patients harboring the intronic polymorphism, 3 had the -230Y SNP, one the -230Y SNP along with the -261M SNP, and one the -402S SNP. Among 14 patients harboring the S278R polymorphisms, 2 had the -230Y SNP, 1 the -230Y SNP along with the -261M SNP, 1 the -191M SNP, and 1 the -402S SNP (Table 1). 

Furthermore, we analyzed the distribution of the identified *AIRE* gene promoter polymorphisms within the different APS subtypes in the present series of patients (Figure 1) (Table 3). 

Of the 74 enrolled patients, 5 were affected by APS1, 5 by APS2, 47 by APS3, and 17 by APS4 (Figure 1). The heterozygous -230Y SNP was found in 2 out of 5 APS1 patients, one harboring the -261M SNP, the other the -402S SNP (Table 3). As regards the APS2-APS4 types, *AIRE* gene promoter SNPs were present in APS3 and APS4 patients but not in APS2 patients. Indeed, the heterozygous -230Y SNP was found in 12 out of 47 APS3 patients and in 6 out of 17 APS4 patients. The homozygous -230T SNP was found in two APS3 patients. Among APS3 patients, one had the -655R SNP, one the -380S SNP and another one the -191M SNP. Furthermore, we did not detect a clear association of the identified *AIRE* gene promoter polymorphisms with APS3 subtypes.

Notably, the five APS1 patients had received the clinical diagnosis based on their clinical manifestations although not genetically confirmed by *AIRE* gene screening (Table 1). In detail, patient No. 2 (Table 1) had a clinical APECED phenotype since affected by hypoparathyroidism and chronic mucocutaneous candidiasis. The patient was also affected by vernal keratoconjunctivitis caused by excessive allergic inflammation [36]. *AIRE* gene screening revealed the heterozygous loss of function mutation p.Arg203Ter (c.607 C>T, rs755490967) in exon 5 and the heterozygous polymorphism p.Ser278Arg (c.834 C>G, rs1800520) in exon 7 inherited from the mother; furthermore, the heterozygous intronic polymorphism IVS9+6 G>A [35], which could be inherited from both parents, was found. *AIRE* gene promoter analysis showed the presence of heterozygous -230Y and -261M SNPs, inherited from the father. Therefore, in this patient, one allele was affected by exons variants, the other allele by SNPs of the promoter; the final effect of this combination could lead to a reduced AIRE expression, leading to the pathological phenotype.

Remarkably, patient No. 16 (Table 1) was affected by hypoparathyroidism, AD, and secondary ovarian failure. Nevertheless, we did not detect variations through *AIRE* gene screening, but the heterozygous *AIRE* promoter -230Y (C/T) variant was present. The search for the C1858T polymorphism of the *PTPN22* gene was negative. Although whole-exome sequencing studies could help to fully elucidate the contribution of additional genetic risk factors, the heterozygous *AIRE* promoter -230Y (C/T) variant could have contributed to the APECED phenotype of this patient. 

APECED patient No. 74 (Table 1) had all three major APECED symptoms: AD, chronic hypoparathyroidism, and chronic mucocutaneous candidiasis. This female also suffered from atrophic gastritis and primary ovarian failure. *AIRE* gene sequencing revealed the heterozygous polymorphism p.Ser278Arg in exon 7 and two other heterozygous intronic polymorphisms: IVS5+14 C>T (c.652+14 C>T, rs41277546) and IVS13-55 A>G (c.1504-55 A>G, rs41277552). For the IVS5 SNP, there are conflicting interpretations of pathogenicity based on two submissions, as reported in the ClinVar database. Instead, the IVS13 SNP has a benign clinical significance based on one submission, as reported in the same database. Neither *AIRE* gene promoter SNPs nor C1858T polymorphism of the *PTPN22* gene were present. Additional undiscovered new genetic risk factors could have contributed to the pathological phenotype in this patient.

### 2.2. Molecular Modeling of the SNPs of the AIRE Gene Promoter

Secondary structure prediction and lowest folding free energy calculations for the genomic sequence encompassing the sites of the nine variants indicate that compared to the wild type, all variants except for -261M (C/A) (rs934375604) either stabilize or destabilize the DNA structure (Figure 2). The changes in stability associated with the variants might impair the structural organization and any potential functional role presented by these regions.

We examined the conservation of the regions affected by the variants by aligning the genomic sequence of the human and other five mammalian species (Appendix A). We found that most of the variants fall within or near conserved blocks, suggesting that the affected regions may have possible functional roles. We also mapped on the same alignment the transcription factor binding sites, either predicted or confirmed, as reported by Lovewell et al. [34] and found that six out of the nine variants (i.e., -402S (C/G), rs371261300, rs934375604, rs751032, rs184978263, rs1048356976) fall within or near these functional sites (Appendix A). Indeed, owing to the possible changes induced on the three-dimensional DNA structure, the remaining three variants might also affect the transcription factor binding sites. Thus, we propose that impairments in the function of these sites might represent at least partially the pathological mechanism of the variants.

## 3. Discussion

The pathogenesis of complex autoimmunity phenotypes is contributed by SNPs of several susceptibility genes [12]. As can be seen, an altered *AIRE* gene expression causes a functional downstream effect on the transcription of peripheral tissue antigens at the thymus level in perinatal age, and thus, the escape of autoreactive T cells in the bloodstream leads to the occurrence of autoimmunity during postnatal lifetime [7,37,38,39]. In this investigation, we further unravel the possible influence of variations in the *AIRE* gene promoter that could potentially affect *AIRE* expression and entity of its transcriptional activity. We therefore investigated the potential presence of SNPs in the *AIRE* gene promoter in DNA samples from a cohort of 74 patients affected by different APS including APS1 to APS4 [4,5]. As control, a cohort of 81 sex-matched HD was analyzed.

We screened 751bp upstream from the *AIRE* start codon, including *AIRE* minimal promoter for SNPs. As shown in Table 2A, *AIRE* promoter gene polymorphisms identified in APS patients were: the -230Y (C/T), the -230T and the -655R (G/A), which also occurred in HD controls (Table 2B); the -261M (C/A), the -380S (C/G), the -191M (C/A), and the -402S (C/G), which were exclusive of four different patients. Notably, the -402S (C/G) was not previously reported in literature and genome databases, including ENSEMBL and dbSNP (Appendix A).

Notably, molecular modeling studies revealed that all SNPs except for 261M (C/A) (rs934375604) were able to change the stability of nucleic acid structure confirming the possible functional effect of the identified *AIRE* promoter SNPs. As regards the *AIRE* -230Y polymorphism, it is located in a conserved region of the promoter and downstream of, but not within, a reporter ETS-1 (ETS Proto-Oncogene 1) transcription factor binding site and it is known to affect *AIRE* expression [33,34]. It has therefore been suggested that *AIRE* -230Y SNP has the potential to influence the promiscuous gene expression regulated by *AIRE*. In detail luciferase reporter assays demonstrated that the highest *AIRE* promoter activity is determined by the commonest haplotypes *AIRE* -655G *AIRE* -230C, while the lowest is associated with haplotype *AIRE* -655G *AIRE* -230T and detected in 10% of the controls [34]. By screening a cohort of 172 patients with alopecia areata associated with APECED, 4 patients were homozygous for this haplotype, suggesting that *AIRE* -655G *AIRE* -230T could be a susceptibility haplotype for alopecia areata outside APECED; nevertheless, it was pointed out by the authors that this hypothesis should be confirmed by the screening of a larger cohort [34,40]. In our investigation, the *AIRE* -655G *AIRE* -230T associated SNPs were more equally represented in the polyendocrine patients than in controls (Table 2 and Appendix A). Furthermore, even for all the additional *AIRE* promoter SNPs identified in this preliminary investigation, especially for those affecting the fold of nuclei acids, no statistical significance of the frequency in patients versus controls was observed (Appendix A). Therefore, their pathogenetic relevance and significance to autoimmune predisposition remain to be unraveled (*vide infra*).

Remarkably, within the 20 patients presenting the *AIRE* gene promoter -230Y SNP (Table 2A), 2 patients were affected by APS1, 12 by APS3, 6 by APS4 (Table 3). Of note, one patient with APS1 also presented the -261M heterozygous polymorphism (patient n° 2), one patient with APS3B the -655R heterozygous polymorphism (patient n° 26) and one patient with APS3A the -380S heterozygous polymorphism (patient n° 29). The -230T homozygous SNP was detected in one patient (n° 67) affected by APS3A and in one patient (n° 36) affected by APS3B (Table 1). Considering the other two identified *AIRE* gene promoter heterozygous polymorphisms, the -191M and the -402S, they were found in one patient affected by APS3A/3B/3C (n° 51) and in one patient affected by APS1 (n° 59), respectively. Based on these results, a rather similar percentage of APS3 and APS4 patients examined had polymorphisms in *AIRE* gene promoter (36.17% APS3 *versus* 35.29% APS4) (Table 3), suggesting that *AIRE* promoter SNPs could have a role in the pathogenesis of these autoimmune syndromes. Conversely, there was no association between APS2 and *AIRE* promoter polymorphisms (Table 3) although the analysis was carried out in a lower number of APS2 patients (Figure 1). 

In a previous investigation carried out on 158 APECED patients in the Italian territory [41], 10 APS-1 patients had no detectable mutations in the *AIRE* gene in agreement with data obtained from other populations [42,43]. This suggests that not-yet-identified genes could be involved in the development of APS-1, including defects of AIRE partners or of other controllers of promiscuous gene expression. As can be seen, in the present study, we enrolled five patients with clinical APECED phenotypes, although they were not genetically confirmed since the disease is typically caused by *AIRE* gene loss of function mutations either in homozygosity or in compound heterozygosity. *AIRE* gene promoter SNPs were detected in these APS-1 patients (Table 3) suggesting their putative effect on the pathological phenotype. 

Overall, based on the preliminary genetic screening results and the molecular modeling data obtained from this study, it is possible to hypothesize that *AIRE* gene promoter polymorphisms could contribute to autoimmune predisposition in APS patients as previously suggested for patients with alopecia areata [34]. However, future functional studies on cells in vitro and throughout animal models in vivo are necessary to validate this hypothesis and thus the actual contribution of *AIRE* gene promoter variants on *AIRE* gene expression. Finally, further extensive genetic screening of *AIRE* gene promoter polymorphisms should be undertaken in larger cohorts of APS patients to validate the effect of an altered *AIRE* gene transcription activity in addition to AIRE deficiency at the thymus level.

Based on the results of future screenings on an extended population of APS patients we could verify whether the distribution of the identified SNPs is selective in the different APS categories that present peculiar associations of autoimmune manifestations. We need to point out that, as reported in the Introduction, autoimmune polyglandular syndromes are complex for their clinical manifestations but also for their causative genetic background. SNPs of the *AIRE* gene promoter, at least those that in the molecular modeling analysis demonstrated being able to affect the structure of nucleic acids, could contribute to the pathogenesis in combination with SNPs of other discovered or not yet discovered susceptibility immune regulatory genes. As final remark, the results of an extended analysis could eventually allow to evaluate the presence of particular SNPs of the *AIRE* gene promoter and the response to the combined treatments that patients receive for the management of the clinical manifestations of each APS syndrome. These potential results could indeed have translational significance in clinical practice.

## 4. Materials and Methods

### 4.1. Study Population

In total, 74 patients affected by APS, i.e., variable association of organ and non-organ specific autoimmune disorders (24 males, 50 females with age ranges at presentation between 0.9 and 19.6 years old) were recruited from the University Department of Pediatrics (DPUO) at Bambino Gesù Children’s Hospital (OPBG) in Rome (Table 1). According to the current criteria for classification of APS [4,5,6,44,45], five patients were affected by clinical APS1 (patients No. 2, 16, 59, 63, 74) not confirmed by the detection of homozygous or compound heterozygous *AIRE* gene mutations. Additionally, 5 patients were affected by APS2 (patients n° 15, 25, 32, 48, 64), 47 patients were affected by APS3 (patients No. 1, 3, 4, 5, 6, 8, 9, 10, 11, 12, 13, 14, 17, 18, 19, 20, 21, 22, 23, 26, 27, 29, 33, 36, 37, 39, 40, 41, 42, 44, 46, 47, 49, 51, 52, 53, 54, 55, 57, 58, 65, 66, 67, 68, 70, 71, 72), and 17 patients were affected by APS4 (patients No. 7, 24, 28, 30, 31, 34, 35, 38, 43, 45, 50, 56, 60, 61, 62, 69, 73). Of the overall series, 35 patients could be classified as APS3A (patients No. 1, 3, 4, 5, 8, 12, 13, 14, 18, 19, 20, 21, 22, 23, 27, 29, 33, 37, 39, 40, 41, 42, 44, 46, 47, 49, 51, 52, 58, 65, 66, 67, 68, 70, 72), 23 patients could be classified as APS3B (patients No. 6, 8, 9, 10, 12, 13, 14, 18, 19, 20, 26, 33, 36, 37, 41, 46, 47, 49, 51, 57, 58, 68, 71), 14 patients could be classified by APS3C (patients No. 1, 10, 11, 17, 44, 47, 49, 51, 53, 54, 55, 57, 58, 70), and 3 could be classified as APS3D (patients No. 11, 13, 58).

Informed consent was obtained from all those who took part in the present study in accordance with the Declaration of Helsinki. The investigation was approved by the local Institutional Review Board (IRB) of Bambino Gesù Children’s Hospital (OPBG), which regulates human samples usage for experimental studies (study protocol No. 1385_OPBG_2017). A control group included 81 healthy blood donors (HD). Controls were recruited from the OPBG Blood Transfusion Centre; they had no history of autoimmunity or immunodeficiency, and no autoantibodies were detected in their serum.

### 4.2. Autoantibodies Screening

The patients’ sera were assayed for T1DM-related autoantibodies, i.e., islet cell antibodies (ICA) by immunofluorescence (IFL), glutamic acid decarboxylase isoform 65 Abs (GADAb), tyrosine phosphatase-related islet antigen 2 Abs (IA2Ab), insulin Abs (IAA), and zinc transporter 8 Abs (ZnT8Ab) by enzyme-linked immunosorbent assay (ELISA); for Addison’s disease, i.e., adrenal cortex Abs (ACA) and 21-hydroxylase Abs (21-OHAb) by ELISA; for AITD-related Abs, i.e., thyrotropin (TSH)-receptor Abs (TRAb) by immunoassay (Immulite TSI, Siemens Healthcare, Tarrytown, NY, USA), thyroglobulin Abs (TgAb), and thyroperoxidase Abs (TPOAb) via electrochemiluminescence immunoassay (ECLIA) (Siemens, Erlangen, Germany); for celiac-disease-related Abs by chemiluminescence (ADVIA Centaur analyzer, Siemens Healthcare, Germany), i.e., anti-transglutaminase-IgA Abs (TRGAb) and deaminated gliadin-IgG Abs (DGP-IgGAb) by EliA and endomysial Abs (EMA) by indirect immunofluorescence (IFL); for autoimmune hepatic diseases, i.e., anti-liver kidney microsomal Abs (LKMAb), smooth muscle Abs (SMA), liver cytosol type 1 Abs (LC1Ab), and soluble liver antigen Abs (SLAIgG); and for stomach-related Abs i.e parietal cells Abs (PCA) by IFL. Intrinsic factor Abs (IFIAb) were tested by ELISA. Non-organ-specific Abs, i.e., nuclear Abs (ANA), extractable nuclear antigen (ENA) (ELiA, Thermo Fisher, Waltham, MA, USA), neutrophil cytoplasmic Abs (ANCA), double-stranded DNA Abs (dsDNAAb), centromere Abs (CeAb), SCL-70 antigen Abs (SCL-70Ab), reticulin Abs (ARA), mitochondrial Abs (AMA), ribosomal Abs (RAb), phospholipid Abs or anti-cardiolipin Abs (CAb), beta2glycoprotein I-IgG or IgM Abs (β2GP-1IgGAb), anti-beta2glycoprotein I-IgM (β2GP-1IgMAb), dense fine speckled 70 Abs (DSF70Ab) (ELiA), and cyclic citrullinated peptide Abs (CCPAb) (EliA) were also tested.

### 4.3. Molecular Studies

To study the *AIRE* gene sequence, the *AIRE* promoter sequence, and the *PTPN22* gene sequence, leukocyte genomic DNA was extracted from whole-blood samples of patients by QIAmp DNA blood mini kit (Qiagen, Hilden Germany) according to the manufacturer’s guidelines.

#### 4.3.1. AIRE Gene Screening

All 14 exons and flanking exon-intron boundaries of the *AIRE* gene (GenBank ID: 326) were sequenced according to already described protocols (Genetic Analyzer 3500 Applied Biosystems HITACHI system, Thermo Fisher Scientific, Rodano, Italy) in the DNA of recruited patients [35]. *AIRE* gene promoter was screened by polymerase chain reaction (PCR) using the following primer sequences: forward 5′-GGAACCGAGGCTCAGAGAAGG-3′ and reverse 5′-CCTCAGAAGCCGGCGTAGC-3′ (annealing temperature 62 °C). These primers are positioned 751bp upstream and 33bp downstream relative to the *AIRE* start codon. The amplification lasted 35 cycles, generating PCR products of 787bp, which were purified using a NucleoSpin Gel and PCR Clean-up kit (Macherey-Nagel, Dueren, Germany) and sequenced with the Genetic Analyzer 3500 (Applied Biosystems HITACHI system).

#### 4.3.2. Screening for the Presence of C1858T PTPN22

Detection of the C1858T variant in the *PTPN22* gene was carried out by PCR with specific primers for exon 14 of the *PTPN22* gene (protein tyrosine phosphatase N22, GenBank ID: 26191): forward 5′-GATAATGTTGCTTCAACGGAATTT-3′ and reverse 5′-CCTCAAACTCAAGGCTCACAC-3′ (annealing temperature 58.5 °C). The amplification lasted 35 cycles, generating PCR products of 318bp that were purified using NucleoSpin Gel and PCR Clean-up kit (Bioanalysis). PCR sequencing was carried out with the BigDye Terminator v.3.1 Cycle sequencing protocol (Life Technologies, Applied Biosystems, Paisley, UK), as reported [18,46].

### 4.4. Multiple Sequence Alignment and Modelling of DNA Secondary Structure

Multiple sequence alignment was performed with MAFFT (v7.490) [47]. Secondary structure predictions were carried out with RNAstructure (v6.4) [48] on the human sequence (ENST00000291582.6 for the wild type and for each variant by introducing the respective nucleotide change) in the range from 1kb upstream exon 1 to 144 bases downstream exon 2.

## Figures and Tables

**Figure 1 ijms-25-02656-f001:**
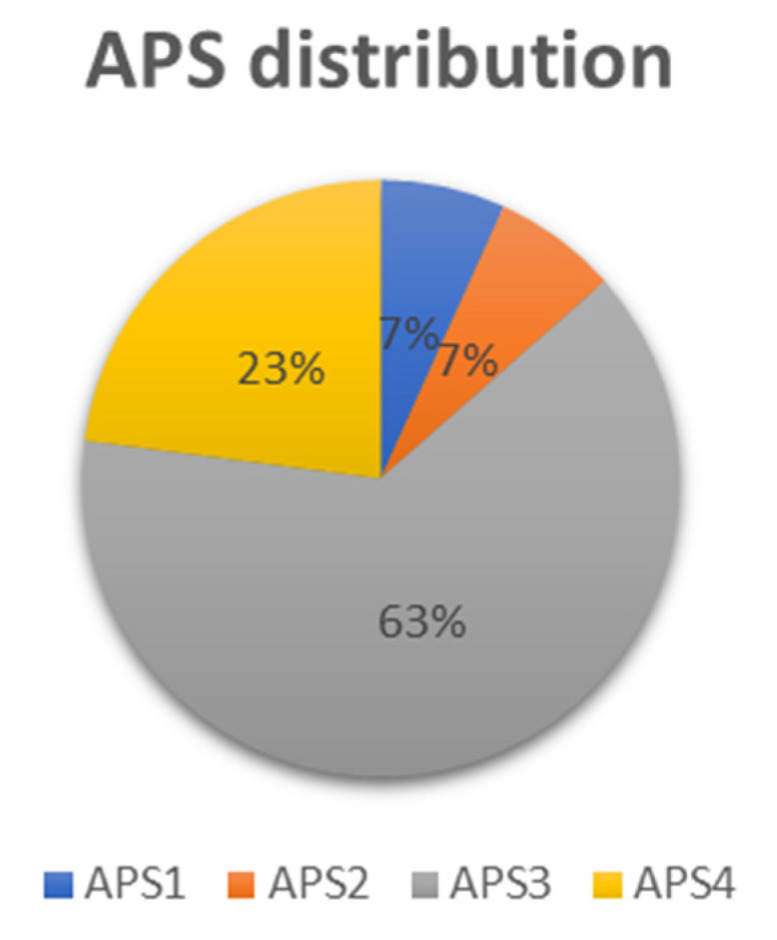
Frequency of APS subtypes in the present patient’s series. Total patients with APS1: 5 (blue); total patients with APS2: 5 (orange); total patients with APS3: 47 (gray); total patients with APS4: 17 (yellow).

**Figure 2 ijms-25-02656-f002:**
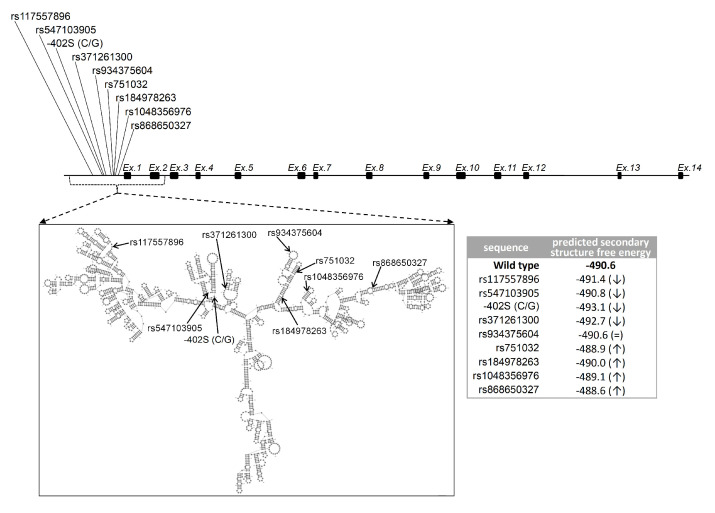
Scheme of *AIRE* gene and DNA secondary structure predictions. Shown is the genomic sequence scheme of *AIRE* showing the positions of exons/introns, the sites of the variants, and a representation of the lowest free energy DNA secondary structure (modelling was made across the nucleotide range indicated by lines). The lowest folding free energies for the wild type and the variants are displayed in the table (the arrows ↑ and ↓ flanking the variant energy values respectively indicate structural destabilization and stabilization with respect to the wild type *AIRE*).

**Table 1 ijms-25-02656-t001:** Clinical and genetic characteristics of the 74 APS patients.

Pt n°	Gender	Age (yrs)	Age at Referral (yrs)	APS Type	Clinical Manifestations	*AIRE* Promoter SNP	Therapy
1	F	26.3	4.4	3A, 3C	T1DM, HT, vitiligo		Insulin aspart, insulin degludec
2	F	3.8	1.4	1	CMC, HP, Vernal keratoconjunctivitis	het -230Y (C/T) het -261M (C/A)	Ketotifen fumarate eye drops, calcitriol
3	F	33.2	12.7	3A	T1DM, HT		Insulin
4	M	28.6	8	3A	T1DM, HT		Insulin aspart, insulin degludec
5	M	40.3	14.2	3A	T1DM, HT		Insulin, bisoprolol fumarate, ramipril
6	F	17.5	4.4	3B	HT, CD	het -230Y (C/T)	None
7	F	2.6	2.5	4	AA, autoimmune piastrinopenia		Topical therapy
8	F	11.9	6.2	3A, 3B	T1DM, HT, CD, AIH		Prednisone, AZA, VitD, insulin lispro
9	F	22.6	5.9	3B	HT, AIH		Prednisone, AZA, UDCA, LT4, VitD
10	F	18	3.9	3B, 3C	HT, suspected CD, AA, allergic rhinitis	het -230Y (C/T)	None
11	F	19.3	6.2	3C, 3D	HT, AA, onychodystrophy, allergic rhinitis, arthralgia, Raynaud phenomenon, recurrent infections		LT4, VitD
12	F	12.1	5.2	3A, 3B	T1DM, HT, AIH		AZA, methimazole, VitD, insulin lispro
13	F	20	11.7	3A, 3B, 3D	T1DM, HT, psoriasis, arthralgia, synovitis		LT4
14	F	29.5	8.8	3A, 3B	T1DM, HT, CD	het -230Y (C/T)	Insulin
15	F	12.7	8	2	HT, AD, IGTT		LT4, hydrocortisone, acetate fludrocortisone, VitD, iron pidolate
16	F	NA	NA	1	AD, HP, secondary ovarian failure	het -230Y (C/T)	NA
17	M	19.2	11.5	3C	Preclinical T1DM, HT, AA		None
18	F	38.7	14.8	3A, 3B	T1DM, HT, CD		LT4, insulin glulisine, insulin glargine
19	M	25.1	2.8	3A, 3B	T1D, HT, CD	het -230Y (C/T)	Insulin
20	F	33.2	9	3A, 3B	T1DM, Basedow’s disease, POF		Insulin aspart, VitD, candesartan cilexetil, methimazole, propranolol hydrochloride, folic acid
21	M	35.3	3.9	3A	T1DM, HT		Insulin aspart
22	F	22.5	0.9	3A	T1DM, HT		Insulin aspart, Insulin glargine
23	M	38.4	16.6	3A	T1DM, HT	het -230Y (C/T)	Human insulin, Insulin detemir, LT4
24	M	29.8	8.3	4	T1DM, CD	het -230Y(C/T)	Human insulin, Insulin detemir, Insulin glulisine
25	F	20	NA	2	HT, AD, psoriasis		NA
26	M	14.2	12.8	3B	HT, AIH	het -655R (A/G) het -230Y (C/T)	Cyclosporine, AZA, UDCA, LT4
27	M	29.8	13.5	3A	T1DM, HT		Insulin aspart, Insulin degludec
28	F	25.1	3.8	4	T1D, CD, vitiligo	het -230Y (C/T)	Insulin aspart
29	F	32.1	5	3A	T1DM, HT	het -380S (C/G) het -230Y (C/T)	Insulin aspart, LT4
30	M	39.9	19.6	4	T1DM, CD		Insulin aspart
31	F	15.8	1.8	4	T1DM, CD, AIH, IgA deficit, obesity		Metformin, insulin glargine, insulin aspart, insulin lispro
32	F	22	8.1	2	HT, CD, AD		Hydrocortisone, fludrocortisone, LT4
33	F	25.4	4.7	3A, 3B	T1DM, HT, suspected CD		None
34	M	25.7	5.1	4	T1DM, CD	het -230Y (C/T)	Human insulin, insulin glargine
35	M	NA	NA	4	T1DM, CD		NA
36	F	20.2	NA	3B	HT, CD, type 2 AIH	hom -230T	NA
37	M	36.5	12.6	3A, 3B	T1DM, preclinical HT, CD	het -230Y (C/T)	Insulin aspart
38	M	34.4	12.7	4	T1DM, CD		Insulin glulisine, insulin glargine
39	M	30.6	9.5	3A	T1DM, HT	het -230Y (C/T)	Insulin aspart, insulin degludec
40	F	29.2	5.4	3A	T1DM, HT	het -230Y (C/T)	Insulin aspart, insulin glargine
41	F	30	2.2	3A, 3B	T1DM, HT, CD		LT4, insulin aspart, insulin glargine
42	F	35	15	3A	T1DM, HT		LT4, insulin aspart, insulin degludec
43	F	23.8	2.8	4	T1DM, CD		Insulin aspart, insulin degludec
44	F	29.7	9.2	3A, 3C	T1DM, HT, vitiligo		Insulin aspart
45	M	28.2	6.5	4	T1DM, autoimmune hemolytic anemia	het -230Y (C/T)	Insulin aspart
46	M	33	13.8	3A, 3B	T1DM, HT, AG	het -230Y (C/T)	LT4, Insulin aspart, insulin degludec
47	F	33.7	10.5	3A, 3B, 3C	T1DM, HT, vitiligo, AG		LT4, Insulin aspart, insulin glargine
48	F	13.2	NA	2	preclinical T1DM, AD		NA
49	F	32.2	6.8	3A, 3B, 3C	T1DM, HT, AG, iron deficiency anemia		Insulin aspart
50	F	28.6	11.2	4	T1DM, CD	het -230Y (C/T)	None
51	M	29.4	3.1	3A, 3B, 3C	T1DM, HT, CD, vitiligo, AG	het -191M (C/A)	Insulin aspart, insulin glargine, LT4, VitB12
52	F	33.4	8.2	3A	T1DM, HT		Insulin aspart, LT4
53	M	22.2	7.8	3C	HT, vitiligo, inhalant allergy, selective IgA deficiency		Betacarotene
54	M	NA	NA	3C	HT, vitiligo		NA
55	F	15.2	1	3C	preclinical HT, AA		Folic acid
56	M	7.1	6.7	4	AD, CMC, AA, arthritis, Perthes’ disease		Hydrocortisone, fludrocortisone, ibuprofen, omeprazole
57	M	8.8	4.7	3B, 3C	HT, CD, AA, atopy		Calcifediol, folic acid, LT4, betamethasone dipropionate, resorcin, levocystin
58	F	13.9	5.9	3A, 3B, 3C, 3D	T1DM, HT, CD, AG, arthritis, vasculitis, algodystrophy, autoimmune encephalitis, ALPS	het -230Y (C/T)	Insulin, sirolimus, methotrexate, micronized PEA, 5-HTP, gabapentin
59	F	13.3	NA	1	T1DM, HT, CMC	het -402S (C/G)	NA
60	M	16	1.9	4	AD, Blackfan-Diamond anemia	het -230Y (C/T)	Fludrocortisone, hydrocortisone, deferasirox, folic acid
61	F	16.4	1.1	4	Preclinical CD, AD, MERS, MIS-C, hyperpigmentation, chronic asthenia		Omeprazole, hydrocortisone, fludrocortisone, topiramate
62	F	17.2	NA	4	CD, HP		NA
63	F	5.9	NA	1	Vitiligo, CMC		NA
64	M	23.8	4.1	2	T1DM, HT, preclinical AD, GH deficit, autoimmune leucopenia		Insulin aspart
65	F	30.9	3.2	3A	T1DM, preclinical HT		Insulin aspart, Insulin detemir
66	F	N.A	NA	3A	T1DM, HT		NA
67	F	NA	NA	3A	T1DM, HT	hom -230T	NA
68	F	26.4	4.7	3A, 3B	T1DM, preclinical HT, preclinical CD		Insulin aspart, Insulin degludec
69	F	26.9	5.2	4	T1DM, CD		NA
70	F	NA	NA	3A, 3C	T1DM, HT, vitiligo		NA
71	F	31.7	11.2	3B	HT, CD		LT4
72	F	29.8	6	3A	T1DM, HT		LT4, insulin aspart, insulin glargine
73	F	30	0.6	4	T1DM, CD		Insulin lispro, insulin degludec
74	F	NA	NA	1	AD, CMC, AG, HP, POF		NA

Pt, patients; Yrs, years; het, heterozygous; hom, homozygous; AD, Addison’s disease; AG, atrophic gastritis; AIH, autoimmune hepatitis; ALPS, autoimmune lymphoproliferative syndrome; CD, celiac disease; CMC, chronic mucocutaneous candidiasis; GH, growth hormone; HT, Hashimoto thyroiditis; HP, hypoparathyroidism; IGTT, impaired glucose tolerant test; MERS, mild encephalitis/encephalopathy with reversible splenial lesion; MIS-C, multisystem inflammatory syndrome in children; POF, primary ovarian failure; T1DM, type 1 diabetes. 5-HTP, L-5-hydroxytryptophan; AZA, azathioprine; LT4, levothyroxine; PEA, palmitoylethanolamide; UDCA, ursodeoxycholic acid; VitD, vitamin D; VitB12, vitamin B12. NA, not available.

**Table 2 ijms-25-02656-t002:** Genotypes and alleles frequency of identified *AIRE* promoter variants in (**A**) 74 patients and (**B**) 81 controls of the present series.

(A)					
GENOTYPES	N°	%	ALLELES	N°	%
-230Y (C/T)	20	27.0	-230C	124	83.8
-230T	2	2.7	-230T	24	16.2
-655R (G/A)	1	1.4	-655G	147	99.3
-655A	0	0.0	-655A	1	0.7
-655G -230Y	19	25.7	-261C	147	99.3
-655G -230T	2	2.7	-261A	1	0.7
-261M (C/A)	1	1.4	-380C	147	99.3
-261A	0	0	-380G	1	0.7
-380S (C/G)	1	1.4	-191C	147	99.3
-380G	0	0.0	-191A	1	0.7
-191M (C/A)	1	1.4	-402C	147	99.3
-191A	0	0.0	-402G	1	0.7
-402S (C/G)	1	1.4			
-402G	0	0			
**(B)**					
**GENOTYPES**	**N°**	**%**	**ALLELES**	**N°**	**%**
-230Y (C/T)	19	23.5	-230C	137	84.6
-230T	3	3.7	-230T	25	15.4
-655R (G/A)	8	9.9	-655G	154	95.1
-655A	0	0.0	-655A	8	4.9
-655G -230Y	18	22.2	-452C	161	99.4
-655G -230T	3	3.7	-452T	1	0.6
-452Y (C/T)	1	1.2	-214C	161	99.4
-452T	0	0.0	-214A	1	0.6
-214M (C/A)	1	1.2	-124C	161	99.4
-214A	0	0.0	-124A	1	0.6
-124M (C/A)	1	1.2			
-124A	0	0.0			

**Table 3 ijms-25-02656-t003:** Distribution of *AIRE* promoter SNPs according to APS subtypes.

*AIRE* Promoter SNP	APS1	APS2	APS3	APS4
-230Y (C/T)	2/5	0/5	12/47	6/17
-230T	0/5	0/5	2/47	0/17
-655R (A/G)	0/5	0/5	1/47	0/17
-261M (C/A)	1/5	0/5	0/47	0/17
-380S (C/G)	0/5	0/5	1/47	0/17
-191M (C/A)	0/5	0/5	1/47	0/17
-402S (C/G)	1/5	0/5	0/47	0/17

## Data Availability

The data presented in this study are available on request from the corresponding author and clinician Doctor Marco Cappa.

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
