# Peer review of "Analysis of the AIRE Gene Promoter in Patients Affected by Autoimmune Polyendocrine Syndromes"

_ijms, 2024, doi:10.3390/ijms25052656_

Round 1

Reviewer 1 Report

Comments and Suggestions for Authors

In the present manuscript, the authors study in patients with APS the presence of single nucleotide polymorphisms (SNP) in the promoter of the gene that regulates the AIRE transcription factor.The authors describe six SNPs, some of them not previously described.

The work is well presented methodologically, but although the authors do not explicitly mention that the described SNPs provide susceptibility for the development of APS, there are some aspects that must be clarified.

First and most importantly, the authors do not perform any statistics comparing the presence of SNPs between patients and healthy controls. This item is essential to know if the SNPs cause susceptibility to the disease.Due to the number of patients and controls included in the study, a Kruskall Wallis test could be performed and would be very clarifying about the effect of the SNPs.

On this same issue, the authors state in the discussion (line 238) that the presence of the polymorphisms in the healthy controls indicates that the controls that express the SNPs have a greater susceptibility to suffering from APS.This statement is not true without knowing the influence of SNPs on the development of the APS.

There are other aspects of the manuscript that should be improved.

1.- The introduction is excessively long and should be focused exclusively on aspects related to work.

2.- The acronym AITD is not defined.

3.- Table 1 is excessively long. The authors should make an effort to present it in a more summarized way.

4.-Table 3B is confusing. An explanation of the table should be provided that clarifies what the table presents.

Author Response

In the present manuscript, the authors study in patients with APS the presence of single nucleotide polymorphisms (SNP) in the promoter of the gene that regulates the AIRE transcription factor.The authors describe six SNPs, some of them not previously described.

The work is well presented methodologically, but although the authors do not explicitly mention that the described SNPs provide susceptibility for the development of APS, there are some aspects that must be clarified.

First and most importantly, the authors do not perform any statistics comparing the presence of SNPs between patients and healthy controls. This item is essential to know if the SNPs cause susceptibility to the disease. Due to the number of patients and controls included in the study, a Kruskall Wallis test could be performed and would be very clarifying about the effect of the SNPs.

We acknowledge Reviewer 1 for this comment. In the statistical analysis we carried out the variables analyzed are dichotomous, presence or absence for each expression and the comparisons are two by two per group (patients and controls). The Kruskall Wallis test was not applicable because it is based on a quantitative distribution of values and number of groups >2. The most appropriate test to compare the distribution of these proportions was chi-square or Fisher's exact test depending on the number of frequencies.

Comparison was performed to evaluate the presence of SNPs between patients and healthy controls. The only statistically significant difference between patients and controls was for the -655R frequency, the presence of this SNP was equal to 11%  (patients) vs 89% (controls) in the two groups respectively, p=0.024.

Results of the analysis are reported in the Supplementary materials as Supplementary S3 Table. See also modified text of the Discussion from line 262 in the revised version of the manuscript with track changes.

As specified in the manuscript since the original version the issue on whether the identified AIRE gene promoter polymorphisms could contribute to APS predisposition, their frequency, mostly for those that are demonstrated in in silico studies to change energy/folding structure of nuclei acids, should be validated by screening larger patients cohorts and healthy controls and in vitro and in vivo functional studies.

On this same issue, the authors state in the discussion (line 238) that the presence of the polymorphisms in the healthy controls indicates that the controls that express the SNPs have a greater susceptibility to suffering from APS. This statement is not true without knowing the influence of SNPs on the development of the APS.

This sentence was amended accordingly in the revised version of the manuscript with track changes from line 260

There are other aspects of the manuscript that should be improved.

1.- The introduction is excessively long and should be focused exclusively on aspects related to work.

The Introduction was significantly shortened by focusing on background informations related to the investigation (see also reply to Reviewer 2 point 1).

2.- The acronym AITD is not defined.

This was specified in the Introduction as ‘autoimmune thyroid disease’ as first mentioned.

3.- Table 1 is excessively long. The authors should make an effort to present it in a more summarized way.

We summarized important clinical ad genetic features in Table 1 which is considerably shortened in the revised version of the manuscript. The additional data were reported in the Supplementary Table S2 (see also reply to Reviewer 2 point 2).

4.-Table 3B is confusing. An explanation of the table should be provided that clarifies what the table presents.

Table 3B is a detailed representation of the AIRE promoter SNPs of the APS3 patients observed in Table 3A within the different APS3 subgroups where patients fall according to the criteria of the most recent classification of APS3 reported in reference 5 (Betterle C. doi: 10.1007/s40618-022-01994-1).

Considering the limited number of patients analysed and the absence of a specific correlation of peculiar SNPs to a particular category of APS3 patients, the information provided is redundant therefore we removed Table 3B from the revised version of the manuscript. ‘Data not shown’ was added on page 9 line 170 of the revised version of the manuscript with track changes.

Reviewer 2 Report

Comments and Suggestions for Authors

The main question addressed by the research is the potential association between AIRE gene promoter SNPs and susceptibility to autoimmune polyglandular syndromes (APS).

The authors found 6 SNPs (-230Y, -655R, -261M, - 380S, -191M, -402S) of the AIRE promoter in APS patients’ DNA. Molecular modeling studies highlighted that all identified SNPs, except for -261M, were able to change energy/folding structure of nucleic acids.

The conclusions are consistent with the evidence and arguments presented. AIRE promoter polymorphisms could contribute to APS predisposition.

Although this is an interesting paper, I find it lacking overall. The outlook is also poor.

1. The introduction is sufficient, but it doesn't connect to the content that follows, and I find it an anticlimax.

2. Table 1 is too long. Please summarize the important points and put the rest in an additional table.

3. Why not compare the clinical features of the different major genotypes or alleles?

4. The discussion is short. For example, don't the authors consider SNPs and treatment response? Investigation of the association between SNPs and clinical features in APS may be considered.

5. How could the results of this research be used in clinical practice in the future?

Author Response

The main question addressed by the research is the potential association between AIRE gene promoter SNPs and susceptibility to autoimmune polyglandular syndromes (APS).

The authors found 6 SNPs (-230Y, -655R, -261M, - 380S, -191M, -402S) of the AIRE promoter in APS patients’ DNA. Molecular modeling studies highlighted that all identified SNPs, except for -261M, were able to change energy/folding structure of nucleic acids.

The conclusions are consistent with the evidence and arguments presented. AIRE promoter polymorphisms could contribute to APS predisposition.

Although this is an interesting paper, I find it lacking overall. The outlook is also poor.

  1. The introduction is sufficient, but it doesn't connect to the content that follows, and I find it an anticlimax.

The Introduction was improved in the revised version of the manuscript also following the comment raised by Reviewer 1 and deleting detailed informations on SNPs of non-HLA genes to make the background of the study more directly focused to the aim expressed in the last paragraph of the Introduction.

  1. Table 1 is too long. Please summarize the important points and put the rest in an additional table.

We summarized important clinical ad genetic features in Table 1 which is considerably shortened in the revised version of the manuscript. The additional data were reported in The Supplementary Table S2 (see also reply to Reviewer 1 point 3).

  1. Why not compare the clinical features of the different major genotypes or alleles?

We acknowledge this reviewer for this comment. Although this investigation is preliminary and should be extended to a higher number of APS patients, indeed  as reported in Table 3A, we specified the occurrence and correlation of the identified SNPs of the AIRE gene promoter within the 4 different categories of APS patients. Patients are diagnosed as APS 1 to 4 on the basis of specific associations of autoimmune disorders according to the most recent criteria (Betterle C. doi: 10.1007/s40618-022-01994-1) as reported in the Introduction. We wish to remark on page 11/12 lines 267-291 of the revised version of the manuscript with track changes what was already highlighted also in the Discussion in the original version, that SNPs of the AIRE gene promoter mostly are detected in APS3 and APS4 patients, while no SNPs are detected in APS2. SNPs can also be detected in 3/5 APS1 patients that do not harbor the pathogenic mutations of the AIRE gene.

  1. The discussion is short. For example, don't the authors consider SNPs and treatment response? Investigation of the association between SNPs and clinical features in APS may be considered.

We acknowledge this reviewer for this comment. Based on the results of future screenings on an extended population of APS patients we could verify the distribution of the identified SNPs in the different APS categories that present peculiar associations of autoimmune manifestations. We need to point out that, as reported in the Introduction, APS are complex syndromes for their clinical manifestations but also for their genetic background that could be contributed by SNPs of other discovered or not yet discovered susceptibility immune regulatory genes in combination with SNPs of the AIRE gene promoter at least those that in the molecular modeling analysis demonstrated to affect energy/folding structure of nucleic acids.

In an extended analysis we could eventually evaluate the presence of SNPs and the response to the combined treatments that patients receive for the management of symptoms of each APS syndrome.  These considerations are added in the Discussion on page 12 line 302-314 of the revised version with track changes.

  1. How could the results of this research be used in clinical practice in the future?

The results of future extensive screening in the APS population may lead to confirm that peculiar SNPs of the AIRE gene promoter are exclusive of certain APS types, thus  allowing to characterize them in combination with the detection of SNPs of other susceptibility immune regulatory genes. Further there is the possibility that certain SNPs of the AIRE gene promoter could correlate and help to predict the outcome of treatments applied for the management of clinical manifestations in each APS subtype with significant relevance in the clinical practice. These considerations were added at the end of the Discussion.

Round 2

Reviewer 2 Report

Comments and Suggestions for Authors

None.